# Characteristics of Previous Tuberculosis Treatment History in Patients with Treatment Failure and the Impact on Acquired Drug-Resistant Tuberculosis

**DOI:** 10.3390/antibiotics12030598

**Published:** 2023-03-16

**Authors:** Soedarsono Soedarsono, Ni Made Mertaniasih, Tutik Kusmiati, Ariani Permatasari, Wiwik Kurnia Ilahi, Amelia Tantri Anggraeni

**Affiliations:** 1Department of Pulmonology and Respiratory Medicine, Faculty of Medicine, Universitas Airlangga, Surabaya 60131, Indonesia; 2Sub-Pulmonology Department of Internal Medicine, Faculty of Medicine, Hang Tuah University, Surabaya 60244, Indonesia; 3Department of Clinical Microbiology, Faculty of Medicine, Universitas Airlangga, Surabaya 60131, Indonesia; 4Tuberculosis Study Group, Institute of Tropical Disease, Universitas Airlangga, Surabaya 60131, Indonesia; 5Division of Pulmonary Medicine, Ibnu Sina General Hospital, Gresik 61121, Indonesia

**Keywords:** previous tuberculosis treatment, treatment failure, MDR-TB, acquired drug resistant tuberculosis, pharmacokinetic TB drugs

## Abstract

Tuberculosis (TB) treatment failure is a health burden, as the patient remains a source of infection and may lead to the development of multi-drug resistance (MDR). Information from cases of treatment failure that develop into MDR, which is related to a history of previous TB treatment, in accordance with the pharmacokinetic aspect, is one important thing to prevent TB treatment failure and to prevent drug resistance. This was an observational descriptive study in an acquired MDR-TB patient who had a prior history of treatment failure. A structured questionnaire was used to collect information. The questionnaire consisted of a focus on the use of TB drug formulas during the treatment period, as well as when and how to take them. This study included 171 acquired MDR-TB patients from treatment failure cases. An amount of 64 patients received the separated TB drug, and 107 patients received the fixed dose combination (FDC) TB drug. An amount of 21 (32.8%) patients receiving separated TB drug and six (5.6%) patients receiving FDC TB drug took their drug in divided doses. In addition, three (4.7%) patients receiving separated TB drug and eight (7.5%) patients receiving FDC TB drug took their drug with food. An amount of 132 out of 171 (77.2%) patients had a history of incorrect treatment that developed into MDR-TB. Education on how to take the correct medication, both the separate version and the FDC TB drug, according to the pharmacokinetic aspect, is important before starting TB treatment.

## 1. Introduction

Tuberculosis (TB) is a communicable disease that is one of the leading causes of death worldwide. An estimated 10.6 million people hadTB in 2021, an increase of 4.5% from 10.1 million in 2020. Indonesia is rank 2nd, representing 9.2% of its population, which accounts for two-thirds of the global TB cases in 2021 [1]. Drug-resistant TB (DR-TB) continues to be a public health threat [2]. Resistance to rifampicin (RIF), the most effective first-line drug, is of greatest concern. Resistance to RIF and isoniazid (INH) is defined as multidrug-resistant TB (MDR-TB) [3]. Both MDR-TB and rifampicin-resistant TB (RR-TB) require treatment with second-line drugs. Globally, in 2021, there were an estimated 450,000 MDR/RR-TB cases. Indonesia is one of seven countries with the highest burden in terms of numbers of MDR/RR -TB cases, and that accounted for two-thirds of global MDR/RR -TB cases in 2021. The estimated proportion of people with TB who had MDR/RR -TB was 3.6% among new cases and 18% among those previously treated [1]. A previous study in Indonesia reported that 433 MDR-TB patients were from seven (1.7%) new cases, 165 (38%) treatment failures, 160 (37%) relapses, 91 (21%) returns after loss to follow-up, and 10 (2.3%) other cases [4].

Previously treated cases are TB patients who have received one month or more of anti-TB drugs in the past. They are further classified by the outcome of their most recent case of treatment. One of the previously treated TB cases was treatment failure, which is defined as a TB patient whose sputum smear or culture is positive at month five or later after treatment [5,6]. TB treatment failure was reported as the risk factor for development of MDR-TB in a previous study [7]. The treatment for pulmonary TB can be inadequate due to various causes, including an incorrect medication method, which is not in accordance with the aspects of pharmacokinetics [8]. One of the factors reported to be the cause of treatment failure, as well as the occurrence of drug resistance, was a history of non-standard and inadequate TB treatment. Inadequate treatment causes treatment failure and recurrence, which are some of the causes of drug resistance [9,10,11]. First-line TB drugs, such as RIF, INH, ethambutol, and pyrazinamide, based on their pharmacokinetics, are concentration-dependent in activity, where the drug should be taken once a day at the correct dose to obtain the optimal Cmax and Cmax/AUC, which mean that the drug is taken once daily, and not two or three times a day. This type of regiment is called divided dose [12,13]. The interactions between food and drugs could reduce the bioavailability of anti-TB drugs, especially for RIF and INH. TB patients are endorsed to take anti-TB drugs at fasting condition to avoid therapeutic failure due to reduced blood concentrations [14]. Poor compliance with Directly Observed Treatment Short-course (DOTS) guidelines and inadequate care delivery that result in treatment failure and relapse are major causes of drug resistance in tuberculosis [15]. DOTS is a specific strategy, endorsed by the World Health Organization (WHO), to improve adherence by requiring health workers, community volunteers, or family members to observe and record patients taking each dose [16,17]. The main aspect of the DOTS strategy is direct supervision of the process of taking medication as it relates to drugs that are always available. Therefore, the adherence is guaranteed when taking TB drugs during the treatment period. In Indonesia, medication supervision is conducted by patients’ family member who have been given education concerning when the patient will start treatment, and this is not done by a nurse or technician [18]. However, although the drugs are taken regularly, the administration of anti-TB drugs, without considering the aspects of pharmacokinetics, can lead to treatment failure and drug resistance. Based on pharmacokinetics aspect, there were two types of drugs available for TB treatment in the community: the separated drug and the FDC TB drug formula. FDC was a drug package that contains certain active drug components [19]. The formulas of FDC TB drugs are listed as RIF, INH, pyrazinamide, and ethambutol. They are called a 2 FDC TB drug if they contain RIF and INH, and they are called 3 FDC if they contain RIF, INH, and pyrazinamide. They are called 4 FDC if they contains RIF, INH, pyrazinamide, and ethambutol [20]. 4 FDC was given during the first two months during the intensive phase, and 2 FDC was given during four months, following the continuation phase [21]. This study was conducted with adult patients. Hence, the FDC TB drug was meant for these patients, and the FDC TB drug was taken during TB treatment. WHO recommendations adopted by the TB sub-directorate of the Indonesian Ministry of Health use the FDC TB drug in TB treatment services [22], but unfortunately there may not be coordination regarding how many FDC TB drug should be provided and distributed to all health services. In Indonesia, there are 3 level public health facility. Primary health care only provide an initial services and not specialized. Secondary health care provided an initial services and several specialize services. Tertiary health care is a complete specialized services. Primary Health Care in Indonesia is a public health facility who give primary services including TB treatment based on DOTS TB program [23].

Tuberculosis treatment failure is a health burden as the patient remains of infection in the community and it may lead to the development of multidrug resistance [24]. It is important to prevent the emergence and transmission of drug-resistant TB because the second line drugs are less effective, have toxic side effects, and require extended treatment. Moreover, treatment failure subsequently leads to higher mortality rates [25]. To improve treatment outcomes for TB especially in Drug-Sensitive TB (DS-TB), efforts to reduce treatment failure are necessary [26]. Information from Acquired MDR-TB patients from treatment failure cases related to the history characteristics of previous treatment is important for strengthening the TB control program through DOTS to prevent TB treatment failure and drug resistance. This study was conducted to evaluate the characteristic of previous drug-intake history related to how drug-intake TB patients with outcome of treatment failure who were developed drug resistance.

## 2. Results

After selection regarding to flow chart of the inclusion of study subjects, This study included 171 Acquired MDR-TB patients from new TB cases who have treatment failed with first-line standard anti-TB regimen, with mean age of 44 years old from 94 (55%) men and 77 (45%) women. In addition, patients who have comorbid controlled Diabetes Mellitus were 70 of 171 (40.9%). Sites of previous TB treatment were reported in Primary Health Care were 112 (65.5%) patients, hospital were 24 (14%), private clinic were 23 (13.5%), independent general practitioner were 7 (4.1%), and the rest were independent medical specialist were 5 (2.9%) patients. The characteristic of study subjects was presented in Table 1 below.

Table 2 showed there are 2 kind of TB medication formula: Separated TB drug and FDC TB drug.

Patients who received Separated TB drug were 64 (37.4%) and FDC TB drug were 107 (62.6%). In separated drug, patients who taken in divided doses were 21 of 64 (32.8%) and 43 of 64 (67.2%) were taken the Anti TB-drugs all at once. In FDC TB drug, patients who taken in divided doses were 6 of 107 (5.6%) and 101 of 107 (94.4%) were taken the Anti TB-drugs all at once. Mostly, patients who were given anti-TB drug with separated TB drug in divided doses were patients in private clinics amount 11 (17.2%). Also, the most patients who were given anti-TB drug with separated TB drug taken all at once were patients in Primary Health Care amount 24 (37.5%). Furthermore, patients who were given anti-TB drug with FDC TB drug taken in divided doses were patients in Primary Health Care amount 4 (3.7%) and the most patients who were given with FDC TB drug taken all at once were patients in Primary Health Care amount 82 (76.6%).

Table 3 showed patients who received separated TB drug taken with food were 3 of 64 (4.7%) and 61 of 64 (95.3%) were taken ≥2 h before/after food.

For patients who received FDC TB drug, 8 of 107 (7.5%) taken Anti TB-drug with food and 99 of 107 (92.5%) were taken ≥2 h before/after food. The most patients who were given anti-TB drugs with separated TB drug taken with food were patients in private clinics amount 2 (3.1%). Patients who were given anti-TB drugs with separated TB drug taken ≥2 h before/after food were patients in Primary Health Care amount 26 (40.6%) mostly. Moreover, patients who were given anti-TB drugs with FDC TB drug taken with food were patients in Primary Health Care amount 6 (5.6%) and the most patients who were given anti-TB drugs with FDC TB drug taken ≥2 h before/after food were patients in Primary Health Care amount 80 (74.8%).

Table 4 showed education about how to take the medicine by the health worker was commonly reported in 105 (61.4%) patients treated in the Primary Health Care, followed by hospital was 22 (12.9%) patients, private clinic were 21 (12.3%) patients, independent general practitioner were 7 (4.1%) patients, and independent medical specialist were 5 (2.9%) patients. 7 (4.1%) patient treated in the primary healthcare and 2 (1.2%) patients respectively who treated in hospital and private health care did not explained by the health worker about how to take the medicine.

## 3. Discussion

Though since 1994, WHO and the International Union against Tuberculosis and Lung Disease (IUATLD) have recommended the use of FDC TB drug as anti-TB therapy. This is to simplify the therapy, increase the compliance, and prevent the inadvertent medication errors [27]. Separated TB drug only for certain patients such as patients who experience side effects [28]. Most frequent causes correlated with TB resistance included non-implementation of DOTS and other important risk factors are correlated with inadequate drug intake by patients, quality of the drugs, and others [29]. Besides, the acquired resistance could be affected by drug malabsorption [30]. Under standard doses due to the wrong way of drug-intake though regularly will cause treatment failure and continue to become drug-resistance.

This study included 171 Acquired MDR-TB patients who have treatment failed in their previous TB treatments regularly. This study found that the administered of Separated TB drug were 64 patients. This is not in accordance with WHO recommendations that have been adopted in Indonesia. Separated TB drug was given by all the health service sites and the site that provided the most separated drugs was non-Primary Health Care sites that is equal to 38 of 64 (59.4%). In this study, we were not ask about why the drugs given separately, because the questionnaire was only aimed to patients. Overall, every sites who gave the drug separately for patients who taken in divided doses were 21 of 64 (32.8%). This may be because education on how to take medicine through DOTS training or in the medical education curriculum related to TB treatment is not included. Even so, in the Primary Health Care there were still 26 (40.6%) patients who given separated TB drug were 2 (3.1%) patients received in divided doses. In addition, there were 86 patients who received FDC TB drug, but 4 (4.6%) patients received in divided doses. 4 patients who received FDC TB drug in divided doses at Primary Health Care dominated compared to other health service sites. In the countries with high burden of TB where DOTS strategy is not implemented well, TB treatment in private sector is reported as the risk factor for MDR-TB. Failure of anti-TB treatment in the private sector was grouped at moderate risk of resistance and MDR-TB [29]. Indonesia is one of many countries with high burden TB and DOTS strategy is still focus only on Primary Health Care. As an illustration, those who have carried out Drug Sensitive TB (DS-TB) notifications as a component of implementing the DOTS strategy by facility Health Services are Primary Health Care estimated approximately 65%, Public hospital 20%, and private hospitals 15% [31]. In this study, patients who take the medication correctly (FDC TB drug or Separated TB drug in all at once and ≥2 h before/after food) were only 39 of 171 (22.8%) patients.

Of the 64 patients received Separated TB drug, 3 (4.9%) patients taken the medicine with food which given by private clinic and independent medical specialist, while of the 107 patients received FDC TB drug, 8 (7.5%) patients taken the medicine with food which mostly given by Primary Health Care amount 6 (5.6%) and the rest was hospital amount 2 (1.9%). This condition when wrong time medication error can be the cause of resistance in such a way that when the FDC TB drug was taken on divided doses chances of resistance becomes several folds [32]. The interactions between food and drugs could reduce the bioavailability of anti-TB drugs, especially for RIF and INH. TB patients should be endorsed to take anti-TB drugs at fasting condition to avoid therapeutic failure due to reduced blood concentrations [14]. A study in India reported that food lowered anti-TB drug concentrations significantly and delayed absorption [33]. As mentioned above, the quality of the drug is also decisive but in this study, the quality of drug which given to patients were not analyzed.

Other factors may play a role in the occurrence of treatment failure and development of drug resistance. Isoniazid as the key drug of first-line anti-TB treatment, exerts potent early bactericidal activity (EBA). Antimicrobial activity is well-correlated with INH exposure. The wide inter individual variability (IIV) seen in the response to INH could lead to suboptimal concentrations in some patients, resulting in treatment failure and a risk of drug resistance. INH metabolism is also influenced by the genetic polymorphism of N- acetyltransferase 2 (NAT2), which contributes to the high IIV in INH clearance and concentration. NAT2 genetic polymorphisms can be classified into 3 phenotypes: rapid, intermediate, and slow acetylators. The population of Indonesia has a large proportion of slow acetylators. Some NAT2 phenotypes are associated with a higher risk of adverse drug reactions (ADRs) to INH and treatment failure [32]. Rapid acetylator were significantly more frequent in patients with MDR-TB, appropriate dose adjustment of INH is important to prevent treatment failure and acquired drug resistance [34]. The guidelines for using maximal doses INH in Indonesia were 300 mg. In this study, the acetylators status of study subjects were not analyzed.

Another risk factor that impact to failure treatment was MTB strain. This factor could not be neglected especially for some various of MTB strains such as Beijing genotype strains. Previous study reported that MTB Beijing genotype 1.9–3.6 times significantly associated with an increased risk of treatment failure [35]. Study was conducted in Indonesia showed that MTB Beijing genotype strains was less susceptible to TB treatment, though there was no resistance to anti TB drug [36]. This study was not analyzed the genotype strains which may cause failure TB treatment.

Education how to take the drug before starting treatment was important. Medication supervisor has important role for the implementation of DOTS in TB treatment. Unfortunately, WHO guidelines only mentioned that the TB medicine combination should be taken regularly in the same time and didn’t mention about timing exactly according to meal’s time. Questionnare of this study enquired that health services site who did not provide education to patients about how to take TB drug medicine correctly were Primary Health Care was 5 (2.9%) patients, followed by Hospital was 4 (2.3%) patients, and private clinic as well as independent general practitioner amount 1 (0.6%) patient respectively. 161 (94.2%) patients have medication supervisor, while 10 (5.8%) patients have no medication supervisor.

Comorbid conditions associated with malabsorption was also reported to be risk factor for drug resistance [24,29]. In HIV disease, the small intestine is frequently impacted either by opportunistic enteric infections that cause intestinal dysfunction or by the HIV virus itself, which causes malabsorption of the majority of nutrients [37,38]. TB patients with HIV that are commonly with malabsorption have been excluded in this study to avoid biased results. This study found that 42 (24.6%) patients experienced adverse event causing vomiting anti-TB drugs and 70 (40.9%) patients with DM (Table 1). ADRs to first-line anti-TB drugs are common. Moreover, these ADRs result in interruption or revision of the anti-TB treatment that could cause treatment failure and mortality in TB patients [39]. In this study, the occurrence of adverse effects did not lead to discontinuation of treatment. TB patients with uncontrolled DM have a higher risk of treatment failure and TB drug resistance [26,40]. Patients with uncontrolled DM have been excluded in this study to avoid biased results.

In DS TB with impaired kidney function and liver disorders, there needs to be a change in the standard TB regimen. Therefore in this study patients with renal failure or impaired liver function were excluded. The presence of primary mono or polydrug-resistant TB is one of factors associated with treatment failure if only given first-line regimen standard [17]. In this current study, although all study subjects in their previous TB treatment were diagnosed by Xpert MTB/RIF with RIF resistance not detected but this does not rule out the possibility of the presence of primary mono or polydrug-resistant TB. In national TB program adopted from WHO TB guideline, It applies that new TB patients, if GeneXpert results showed RIF sensitive, are immediately given a first-line regimen standard without further confirmation whether there is resistance in other anti-TB drug, especially INH. A previous study in new TB patients with rifampicin susceptible TB (RS-TB) detected by Xpert MTB/RIF reported that INH resistance was detected in 4 (7.4%) using first-line line probe assay (FL-LPA) and 5 (9.3%) using culture-based drug susceptibility test (DST). RIF resistance was also found in 1 (1.9%) using FL-LPA and 2 (3.7%) using culture-based DST [41]. In the subjects of this study, further sensitivity tests were not carried out, especially to INH and also to Ethambutol and Pyrazinamide.

This study has some limitations. It was not all of the TB patients diagnosed initially by microbiological examination (molecular rapid test). The proportion of microbiology-positive and clinical radiology test of TB patients was not evaluated, but all patients diagnosed and treated at health primary care had 100% performed according to regulation, while those at non-Primary Health Care varied (some used molecular rapid test and some did not) and not all the sites implemented DOTS program.

## 4. Materials and Methods

### 4.1. Study Design and Ethical Statement

This was an observational descriptive study in Dr. Soetomo General Academic Hospital Surabaya and Ibnu Sina General Hospital Gresik that was conducted from May 2021 to October 2021. The samples were all Acquired pulmonary MDR-TB patients from new TB cases who have treatment failed with first-line standard anti-TB regimen (4 FDC). Acquired pulmonary MDR-TB patients from new pulmonary TB cases who have treatment failed with first-line standard anti-TB regimen per definition are new case pulmonary TB patients diagnosed based on GeneXpert examination showing positive *Mycobacterium tuberculosis* (MTB) and RIF Resistance not detected, and or positive clinical radiology to TB diagnosis, then given standard treatment regimen first line TB drugs and sputum smear or culture is positive at month 5 or later during treatment as well as GeneXpert examination in those time showed positive MTB RIF resistant detected. For the confirmation, the next examination by culture/sensitivity test forward showed RIF and INH resistant (MDR-TB). New TB cases was defined as TB patients who have never had TB before nor received TB treatment who receive standard treatment but do not recover and instead become MDR TB at the end of the treatment duration.

This study was approved by the ethics committee with ethical clearance number 83/EC/KEPK/FKUA/2022 on 19 May 2022. This study was conducted in accordance with the Declaration of Helsinki. All participants included had given their written informed consent to participate in this study.

### 4.2. Data Collection

A structured questionnaire was used to collect information by interviewing the subjects as respondents who have signed the informed consent. The questionnaire was developed from any validated and previously published articles, and were added from the experiences of physician when providing medical services [10,11,12]. The questionnaire had 22 questions including demographic profile 11 questions and the administration of drug amount 11 questions. Interview was conducted by the physician, peer educators, and patients’ supporters who were trained before collecting data. Completed questionnaires are input by the research assistant and double checked by the investigators. The questionnaire focused on the use of TB drug formulas during the treatment period, when and how to take them, In addition, demographic patients and health service site for treatment. Patients with uncontrolled diabetes mellitus (DM), HIV infection, renal failure, liver disease, and TB patients whose considered to have taken TB medication irregularly were excluded. Data was entered and presented as a table.

Figure 1 is the flow chart of the inclusion of study subjects.

## 5. Conclusions

Acquired MDR-TB from treatment failure history cases of patients in Primary health care contributed wrong administration to take the drug mostly. This study suggest that implementation of DOTS strategy should be expanded to all health service sites and emphasize education on how to take the medicine in accordance with pharmacokinetic aspect. In addition, further research on patients regarding genomic and MTB strains should be conducted as well as testing on other anti-TB drug resistance in new patients whose geneXpert results showed RS-TB and focused on patients who enrolled at health service sites that have implemented DOTS.

## Figures and Tables

**Figure 1 antibiotics-12-00598-f001:**
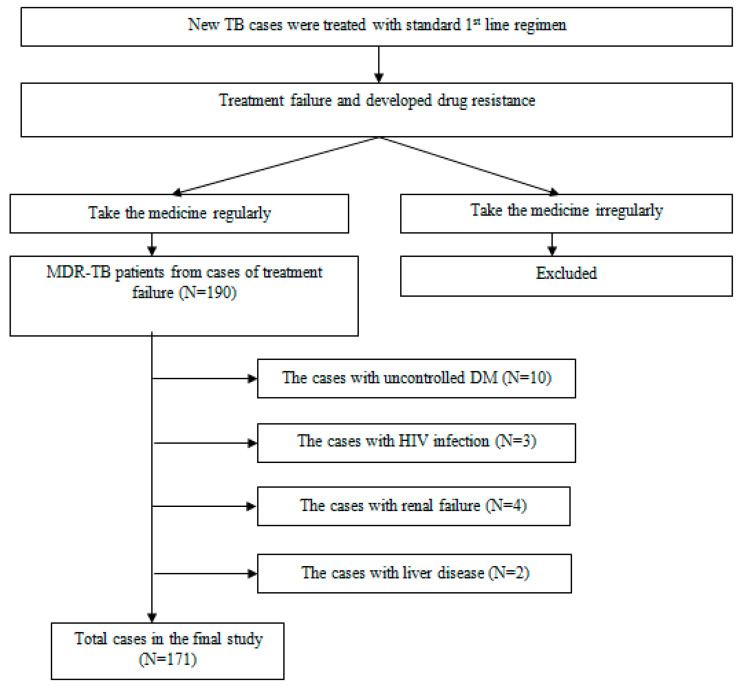
Flow chart of the Inclusion Study Subjects.

**Table 1 antibiotics-12-00598-t001:** Characeristic of Drug-Resistant Tuberculosis Patients from Cases of Treatment Failure with First Line Regimens.

Variable	Total (N = 171)
Age	44 ± 11.9 (19–72)
Mean ± Std. Deviation (Min-Max)	
Sex Men Women	94 (55%) 77 (45%)
Weight (kg)	51.5 ± 9.1 (32–100)
Height (m)	1.6 ± 0.1 (1.4–1.8)
Body Mass Index (kg/m^2^)	20.1 ± 3.3 (12.8–36.7)
Sites of Previous TB Treatment Primary Healthcare Hospital Private Clinic General Practitioner Medical Specialist	112 (65.5%) 24 (14%) 23 (13.5%) 7 (4.1%) 5 (2.9%)
Medication Supervisor (Families)	161 (94.2%)
Adverse Event Causing Vomiting the TB Drugs	42 (24.6%)
Controlled Diabetes Mellitus	70 (40.9%)

**Table 2 antibiotics-12-00598-t002:** Characteristic of Anti-TB drug, How to Take TB Drug, and Sites to Get Medication in the Previous TB Treatment (N = 171).

Anti-TB Drugs	How to Take TB Medicine	Sites of Previous TB Treatment				
		Primary Healthcare (N = 112)	Private Clinic (N = 23)	Hospital (N = 24)	Independent General Practitioner (N = 7)	Independent Medical Specialist (N = 5)
Separated TB drug (N = 64)	Taken in Divided Doses (N = 21)	2 (3.1%)	11 (17.2%)	3 (4.7%)	3 (4.7%)	2 (3.1%)
	Taken All at Once (N = 43)	24 (37.5%)	8 (12.5%)	7 (10.9%)	2 (3.1%)	2 (3.1%)
FDC TB drug (N = 107)	Taken in Divided Doses (N = 6)	4 (3.7%)	1 (0.9%)	1 (0.9%)	0 (0%)	0 (0%)
	Taken All at Once (N = 101)	82 (76.6%)	3 (2.8%)	13 (12.1%)	2 (1.9%)	1 (0.9%)

**Table 3 antibiotics-12-00598-t003:** Characteristic of Anti-TB drugs, Time to Take TB Drug, and Sites to Get TB Medication in the Previous TB Treatment (N = 171).

Anti-TB Drugs	Time to Take TB Medicine	Sites of Previous TB Treatment				
		Primary Healthcare (N = 112)	Private Clinic (N = 23)	Hospital (N = 24)	Independent General Practitioner (N = 7)	Independent Medical Specialist (N = 5)
Separated TB drug (N = 64)	With food (N = 3)	0 (0%)	2 (3.1%)	0 (0%)	0 (0%)	1 (1.6%)
	≥2 h before/after food (N = 61)	26 (40.6%)	17 (26.6%)	10 (15.6%)	5 (7.8%)	3 (4.7%)
FDC TB drug (N = 107)	With food (N = 8)	6 (5.6%)	0 (0%)	2 (1.9%)	0 (0%)	0 (0%)
	≥2 h before/after food (N = 99)	90 (24.3%)	4 (0.9%)	12 (5.6%)	2 (1.9%)	1 (0.9%)

**Table 4 antibiotics-12-00598-t004:** Education How to Take Medicine by Health Worker (N = 171).

Sites of Previous TB Treatment Education	Education How to Take Medicine by Health Worker
Primary Healthcare (N = 112)	105 (61.4%)
Private Clinic (N = 23)	21 (12.3%)
Hospital (N = 24)	22 (12.9%)
Independent General Practitioner (N = 7)	7 (4.1%)
Independent Medical Specialist (N = 5)	5 (2.9%)

## Data Availability

The data presented in this study are available on request from the corresponding author. The data are not publicly available due to patients’ privacy.

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
