# Peer review of "Characteristics of Previous Tuberculosis Treatment History in Patients with Treatment Failure and the Impact on Acquired Drug-Resistant Tuberculosis"

_antibiotics, 2023, doi:10.3390/antibiotics12030598_

Round 1
Reviewer 1 Report
An interesting real-world study, relevant for the understanding ot the reasons why the formal adoption of correct guidelines fails to achieve the goal of an adequate and effective pharmacological treatment against a social disease like TB: but a revision is required.
Some sentences result difficult to comprehend, e.g. in the abstract - lines 23-29.
The "Materials and Methods" section is uncorrectly positioned inside the text.
Necessary to define the Primary Healthcare scenario in Indonesia.
Therefore necessary a second read after corrections.
Author Response
Reply to the reviewers’ comments
We would like to thank the reviewer for all the constructive comments and advice. We respond below in detail to each of the reviewer’s comments. In addition, we refer to the current page to show the revised things. We hope that the reviewers will find our responses to their comments satisfactory.
1 Some sentences result difficult to comprehend, e.g. in the abstract - lines 23-29.
We can't explain it more because of word limit regulation, but we have explained it in the main manuscript (result)
Page 1: Lines 23-29
2 The "Materials and Methods" section is uncorrectly positioned inside the text.
It was based on journal template which managed the section as -abstract -introduction
-result
-discussion
-methods and materials
-conclusion
3 Necessary to define the Primary Healthcare scenario in Indonesia.
In Indonesia, there are 3 level public health facility. Primary health care only provide an initial services and not specialized. Secondary primary health care provided an initial services and several specialize services. Tertiary health care is a complete specialized services. Primary Health Care in Indonesia is a public health facility who give primary services including TB treatment based on DOTS TB program
Page 1: Lines 98-102
4 Therefore necessary a second read after corrections.

Reviewer 2 Report
This study is interesting as it examined the reasons behind the acquisition of MDR-TB in patients with a history of treatment failure of a DS-TB. However, the readability is poor due to language issues. There are also many comments that the authors are recommended to address to improve the manuscript.
Major comments:
Title and abstract:
1. The title is not clear as I had to read it more than once to understand it. I suggest changing it to: Characteristics of Previous Tuberculosis Treatment History in Patients with Treatment Failure and the Impact on Acquired Drug-Resistant Tuberculosis
2. The abstract is longer than the recommended by journal (200 words). Please shorten it or at least remove the headings as it should be unstructured.
3. Line 39: Also add "MDR-TB"
Introduction:
4. Add references to the sentences that end on lines 45, 46, 48, and 50.
5. Line 71: Specify in between parentheses some examples of drugs that interact at the absorption site (e.g., antacids and minerals).
6. Line 74" Describe what DOTS in your country mean. In some countries, it is the delivery of a healthcare provider (a nurse or a technician) of the medication and having the patient taking it in front of them. Does the DOTS involves the family in your country and hence you have the "Supervision of family" in Table 1?
7. Line 84: Spell out "FDC" and list the drugs contained in that formula.
Methods:
8. Line 283: How come they are new but also treatment failed? This means they are not new. If this is true, remove the word "new" to avoid confusion.
9. Line 284: List the 1st line anti-TB drugs in between parentheses.
10. Line 291-292: The sentence is not clear. Do you mean RIF resistance only (you said positive MTB RIF resistant) or RIF and INH resistance? Both are mentioned. Please clarify.
11. How did you identify the patients? From electronic medical records or from the history of GeneXpert results?
12. Line 301: You mentioned that you used previous studies to develop your questionnaire, please cite them at the end of this sentence.
13. Line 301-302: "The questionnaire was filled by interviewing the subjects (acquired MDR-TB patients)." This sentence is a duplicate. You already mentioned that you interviewed them and you already described your participants in the previous section.
14. Specify how many questions did the questionnaire contain. It would also be helpful to include it as a supplementary material.
15. Did you also include patients who were enrolled in the DOTS program? If so, this has to be specified. It should be also listed as a limitation since those patients are already supervised on the way they should take their drugs.
Results:
16. Line 102: Again, the term "new TB cases" is confusing. It implies that those patients are new TB patients whose source of TB was patients with MDR-TB. If this is not correct, then remove "new TB cases"
17. Line 104: Better use the term "controlled diabetes mellitus". Also, did all the included 171 patients have diabetes? If so, then the percentage 40.9% is correct. Otherwise, calculate the percentage only out of the diabetic subgroup of patients. Same should apply to Table 1 (change to "Controlled diabetes mellitus" and add a footnote indicating the actual number of diabetic patients if not all the patients were diabetics.
18. Table 1: What do the authors mean by "Medication supervisor"? Aren't those patients independent adults? Or did they have some sort of disability to need the supervision? Or were they enrolled in the DOT (direct observed therapy) program? Please clarify what this term means in a footnote under the table.
19. Table 1: No need to list the number that corresponds to both Yes and No. Remove the Yes and No and rows and only copy the numbers that correspond to Yes in front of the row title.
20. Lines 115-116: What does the FDC contain? If it's a fixed combination in one pill, how could some patients take it as divided doses? Please clarify.
21. Lines 118-131: This whole part repeats what is already clarified in Table 2. The authors are advised to select only 1-2 major observations to describe them as the reader can find the rest in the table.
22. Table 2: Type the abbreviations below the table (TB and FDC). Also, please define FDC. What drugs does it contain?
23. Lines 134-151: Similar to the previous comment, the authors are suggested to only include the major observations without describing the whole table.
24. Table 3: It looks like there is a miscalculation. The total number under the "Primary Healthcare" column doesn't add up to 112 (adds up to 122). Also, the row of "≥2 h before/after food" doesn't add up to 99 (adds up to 109). Please revise and correct.
Discussion:
25. Lines 193-194: The sentence is a duplicate of the previous one. Please delete.
26. Lines 204-206: Explain why resistance happens. Explain that it is because of impaired absorption due to chelation. Add more details.
27. Lines 210-212: Remove this sentence since this is out of the cope of your work.
28. Lines 237-239: This description of DOT should be moved to the introduction based on the comment made above.
29. Line 248: Give examples on comorbidities that can result in malabsorption.
Conclusion:
30. The conclusion shouldn't summarize the results with the numbers. Simply type why those patients acquired MDR-TB (wrong administration, lack of education, etc. and which were the major sites of those patients with poor treatment education) and then include your recommendations (which you have already mentioned on lines 324-329).
General:
31. The overall readability of the manuscript is low as it contains a lot of language errors and incoherent sentences. The authors are strongly encouraged to consult a language editing service. I noted some of these issues under here are many minor edits that need to be made. Some (but not all) are listed under minor comments below.
Minor comments:
32. Line 25: Add "the" before "pharmacokinetic"
33. Line 26: Make "characteristic" plural
34. Line 27: Remove "were" before "developed"
35. Line 28: Add "study" after "descriptive"
36. Line 28-29: Change it to "who had a prior history of treatment failure"
37. Line 35: Change "during" to "with"
38. Line 111: I think something is written in Indonesian at the end of the table title. Please delete.
39. Methods (line 282): Change "on" to "that was conducted from"
40. Methods (line 303): Add "were" before "trained"
41. Methods (line 305): Remove "consisted of a"
42. Figure 1 seems to have an unusual paragraph symbol. Please remove those symbols.
43. Line 214: Remove "Isoniazid" as you have already defined it as INH in the introduction.
44. Line 274: Start with "This study has some limitations." Then begin listing your limitations.
Author Response
Reply to the reviewers’ comments
We would like to thank the reviewer for all the constructive comments and advice. We respond below in detail to each of the reviewer’s comments. In addition, we refer to the current page to show the revised things. We hope that the reviewers will find our responses to their comments satisfactory.
Title and Abstract
1 The title is not clear as I had to read it more than once to understand it. I suggest changing it to: Characteristics of Previous Tuberculosis Treatment History in Patients with Treatment Failure and the Impact on Acquired Drug-Resistant Tuberculosis
The title of the article has been revised: “Characteristics of Previous Tuberculosis Treatment History in Patients with Treatment Failure and the Impact on Acquired Drug-Resistant Tuberculosis”
Page 1: line 2-3
2 The abstract is longer than the recommended by journal (200 words). Please shorten it or at least remove the headings as it should be unstructured.
The abstract has been shorten and the heading has been removed
Page 2: line 22-38
3 Line 39: Also add "MDR-TB"
“MDR-TB” has been added
Page 2: line 39
Introduction
4 Add references to the sentences that end on lines 45, 46, 48, and 50.
References have been added on lines 45, 46, 48, and 50.
Page 2: -line 45
-line 46
-line 48
-line 50
5 Specify in between parentheses some examples of drugs that interact at the absorption site (e.g., antacids and minerals).
Several drugs gave the interaction when it administered with food such as antacid, mineral, etc including TB drug. But Our manuscript was not mentioned out of TB drug because We think that it is not Our main discussion.
Page 2: line 71
6 Describe what DOTS in your
DOTS is a specific strategy,
Page 2: line
country mean. In some countries, it is the delivery of a healthcare provider (a nurse or a technician) of the medication and having the patient taking it in front of them. Does the DOTS involves the family in your country and hence you have the "Supervision of family" in Table 1?
endorsed by the World Health Organization (WHO), to improve adherence by requiring health workers, community volunteers or family members to observe and record patients taking each dose (16,17). The main of the DOTS strategy is direct supervision of taking medication in addition to drugs that are always available. Therefore, the adherence is guaranteed in taking TB drugs during the treatment period. In Indonesia, medication supervisor is conducted by patients’ family member who have been given education when the patient will start treatment and not by nurse nor technician
76-82
7 Spell out "FDC" and list the drugs contained in that formula.
List of it formula:
-Rifampicin (RIF)
-Isoniazid (INH)
-Pyrazinamide
-Ethambutol
Page 2: line 84
Methods
8 How come they are new but also treatment failed? This means they are not new. If this is true, remove the word "new" to avoid confusion.
The explanation of “new TB cases” has been added because it is important word.
Page 7: line 291
9 List the 1st line anti-TB drugs in between parentheses.
The drug formula were listed.
Page 7: Line 282
10 The sentence is not clear. Do you mean RIF resistance only (you said positive MTB RIF resistant) or RIF and INH resistance? Both are mentioned. Please clarify.
The correct sentences was
Because GeneXpert only showed RIF resistant in rapid moment. Otherwise, Culture/DST showed RIF and INH resistant to prove that it was MDR TB and not only mono-resistant TB.
Narration was repaired to avoid misunderstanding.
Page 7: Line 301
11 How did you identify the patients? From electronic medical records or from the
It comes from electronic medical records
history of GeneXpert results?
12 You mentioned that you used previous studies to develop your questionnaire, please cite them at the end of this sentence.
The citation has been added
Page 8: Line 310
13 "The questionnaire was filled by interviewing the subjects (acquired MDR-TB patients)." This sentence is a duplicate. You already mentioned that you interviewed them and you already described your participants in the previous section. "The questionnaire was filled by interviewing the subjects (acquired MDR-TB patients)." Has been deleted
Page 8: Line 311
14 Specify how many questions did the questionnaire contain. It would also be helpful to include it as a supplementary material.
The listed question for questionnaire has been added as a supplementary material
Supplementary material
15 Did you also include patients who were enrolled in the DOTS program? If so, this has to be specified. It should be also listed as a limitation since those patients are already supervised on the way they should take their drugs
In the Primary Health Care, it must be enrolled as DOTS program but for the other sites, like private clinic, independent general was not enrolled as DOTS program.
Hospital, several patients enrolled as DOTS and the rest were not.
Results
16 Again, the term "new TB cases" is confusing. It implies that those patients are new TB patients whose source of TB was patients with MDR-TB. If this is not correct, then remove "new TB cases"
Yang dimaksud new cases adalah pasien TB yang sebelumnya belum pernah sakit TB atau pernah mendapat pengobatan TB yang kemudian mendapat pengobatan standard namun tidak sembuh dan malah menjadi MDR TB pada akhir durasi pengobatan. Hal ini sudah ditambahkan dalam method.
Page 8: Line 323
17 Better use the term "controlled
“regulated DM” was changed to
Page 3: line
diabetes mellitus". Also, did all the included 171 patients have diabetes? If so, then the percentage 40.9% is correct. Otherwise, calculate the percentage only out of the diabetic subgroup of patients. Same should apply to Table 1 (change to "Controlled diabetes mellitus" and add a footnote indicating the actual number of diabetic patients if not all the patients were diabetics.
“Controlled DM”
The number of controlled DM patients was mentioned as “In addition, patients who have comorbid controlled Diabetes Mellitus were 70 of 171 (40.9%).”
We changed all “regulated” terminology to “uncontrolled”
109
18 What do the authors mean by "Medication supervisor"? Aren't those patients independent adults? Or did they have some sort of disability to need the supervision? Or were they enrolled in the DOT (direct observed therapy) program? Please clarify what this term means in a footnote under the table.
The main of the DOTS strategy is direct supervision of taking medication in addition to drugs that are always available. Therefore, the adherence is guaranteed in taking TB drugs during the treatment period. In Indonesia, medication supervisor is conducted by patients’ family member who have been given education when the patient will start treatment and not by nurse nor technician.
It was added in introduction
Table 1
19 No need to list the number that corresponds to both Yes and No. Remove the Yes and No and rows and only copy the numbers that correspond to Yes in front of the row title.
The Table has been repaired
Table 1
20 What does the FDC contain? If it's a fixed combination in one pill, how could some patients take it as divided doses? Please clarify.
FDC contain was explained in Introduction section.
It means that the patients take it three times a day, and did not take it all the drug at once based on the right pharmacokinetic aspect. For example, based on patients’ weight need 3 tablets a day, it took all at once an did not
Page 3: Line 130
one tablet taken three times a day.
21 This whole part repeats what is already clarified in Table 2. The authors are advised to select only 1-2 major observations to describe them as the reader can find the rest in the table.
The sentences above Table 2 (explanation of Table 2) have been shorten.
Page 4: Line 128-140
22 Type the abbreviations below the table (TB and FDC). Also, please define FDC. What drugs does it contain?
As we explained inIntroduction, List of it formula were :
-Rifampicin (RIF)
-Isoniazid (INH)
-Pyrazinamide
-Ethambutol
(in point 7 )
Table 2
23 Similar to the previous comment, the authors are suggested to only include the major observations without describing the whole table.
The sentences above Table 3 (explanation of Table 3) have been shorten.
Page 4: Line 143-153
24 It looks like there is a miscalculation. The total number under the "Primary Healthcare" column doesn't add up to 112 (adds up to 122). Also, the row of "≥2 h before/after food" doesn't add up to 99 (adds up to 109). Please revise and correct.
It was the correct calculation because the number of all patients was 171 which if We add up all sites the results are 171.
PHC (112) + Private Clinic (23) + Hospital (24) + Independent General Practitioner (7) + IndependenMedical Specialist (5) = 171 patients
Table 3
Discussion
25 The sentence is a duplicate of the previous one. Please delete.
The sentence has been deleted
Page 5: Line 193-194
26 Explain why resistance happens. Explain that it is because of impaired absorption due to chelation. Add more details.
Explanation has been added
Page 5: Line 204-206
27 Remove this sentence since this is out of the cope of your
The sentences has been removed
Page 6: Line 210-212
28 This description of DOT should be moved to the introduction based on the comment made above.
It was explained in Introduction section
Page 7: Line 237-239
29 Give examples on comorbidities that can result in malabsorption
The example has been added
Page 7: Line 245
Conclusion
30 The conclusion shouldn't summarize the results with the numbers. Simply type why those patients acquired MDR-TB (wrong administration, lack of education, etc. and which were the major sites of those patients with poor treatment education) and then include your recommendations (which you have already mentioned on lines 324-329).
The summary has been removed all the numbers and make it simple.
Page 8: Line 316-323
General
31 The overall readability of the manuscript is low as it contains a lot of language errors and incoherent sentences. The authors are strongly encouraged to consult a language editing service. I noted some of these issues under here are many minor edits that need to be made. Some (but not all) are listed under minor comments below.
The English written has been corrected

Round 2
Reviewer 2 Report
Thank you for addressing all the comments appropriately. However, it looks like the response to comment 16 was written in Indonesian instead of English. But I noticed that it was addressed. Authors are advised to review their responses before submitting them to avoid such issues in the future.
The English writing has improved a lot; though, there remain some issues in the parts that are not highlighted, which I think should be edited by the journal's editorial team.